# Ultrahigh-efficient material informatics inverse design of thermal metamaterials for visible-infrared-compatible camouflage

Wang Xi[1,4], Yun-Jo Lee[2,4], Shilv Yu[1], Zihe Chen[1], Junichiro Shiomi[3], Sun-Kyung Kim [2] ✉ & Run Hu [1] ✉

Multispectral camouflage technologies, especially in the most frequently-used visible and infrared (VIS-IR) bands, are in increasing demand for the ever-growing multispectral detection technologies. Nevertheless, the efficient design of proper materials and structures for VIS-IR camouflage is still challenging because of the stringent requirement for selective spectra in a large VIS-IR wavelength range and the increasing demand for flexible color and infrared signal adaptivity. Here, a material-informatics-based inverse design framework is proposed to efficiently design multilayer germanium (Ge) and zinc sulfide (ZnS) metamaterials by evaluating only ~1% of the total candidates. The designed metamaterials exhibit excellent color matching and infrared camouflage performance from different observation angles and temperatures through both simulations and infrared experiments. The present material informatics inverse design framework is highly efficient and can be applied to other multi-objective optimization problems beyond multispectral camouflage.

In nature, numerous organisms, including chameleons, octopuses, and certain insects, possess the remarkable ability to alter their skin coloration for purposes of blending into their surroundings, evading predators, or approaching prey[1,2]. Drawing inspiration from these adaptive mechanisms, humans have sought to conceal target signatures to achieve invisibility and camouflage[3–7], particularly in the contexts of battle fatigue and stealth aircraft. However, the development of detection and sensing technologies has necessitated the creation of antidetection camouflaging techniques tailored to various spectral ranges such as visible, infrared, laser, and microwave. Nevertheless, most camouflage technologies are limited to specific spectral ranges and become ineffective when confronted with multispectral detection and sensing technologies. For example, soldiers wearing battle fatigues can be blended into the background environment, but easily exposed in infrared sensing; stealth aircrafts can be invisible in military radars or infrared detectors but exposed in visible telescope. Therefore, multispectral camouflage technologies, especially visible-infrared (VIS-IR) camouflage, are in high demand for almost ubiquitous surveillance and reconnaissance in our daily lives.

VIS-IR camouflage involves manipulating electromagnetic waves within the visible and mid-infrared wavebands to align the target surface reflectivity or emissivity spectra with desired characteristics. Specifically, the target's reflectivity spectrum within the visible band (0.38–0.78 μm) must exhibit selectivity to generate comparable chromatic values with the background environment[8], thus causing the target to appear visually indistinguishable from the background to human eyes or visible detectors. As for infrared camouflage, the target's emissivity spectrum should be low within the atmospheric window (8–13 μm) to counter the detection of thermal signals by

[1]School of Energy and Power Engineering, Huazhong University of Science and Technology, Wuhan 430074, China. [2]Department of Applied Physics, Kyung Hee University, 1732 Deogyeong-daero, Giheung-gu, Yongin-Si, Gyeonggi-do 17104, Republic of Korea. [3]Department of Mechanical Engineering, University of Tokyo, 7-3-1 Hongo, Bunkyo, Tokyo 113-8656, Japan. [4]These authors contributed equally: Wang Xi, Yun-Jo Lee. ✉e-mail: sunkim@khu.ac.kr; hurun@hust.edu.cn

infrared imagers. Furthermore, outside the atmospheric window, specifically within the 5–8 μm and 13–20 μm ranges, the target's emissivity spectrum should remain high to facilitate radiative cooling and effectively lower the target's temperature. This cooling requirement is particularly crucial for military targets characterized by elevated temperatures[7], such as aircraft tailpipes (~740 K). Nevertheless, meeting these multispectral requirements for achieving VIS-IR camouflage is challenging because regulating the reflected properties in the visible band via structural adjustments will inevitably affect the emissive properties in the infrared bands and vice versa. Moreover, the broad spectral range encompassing hundreds of nanometers (visible band) to tens of microns (infrared band) renders synergetic optimization particularly demanding. To address these challenges, metamaterials[6,9–15], which are artificial structures capable of manipulating electromagnetic waves, have emerged as a promising avenue for achieving VIS-IR camouflage[16,17]. Previous research efforts have achieved visible camouflage and infrared camouflage individually or in a compatible manner using metamaterials, including photo-engineered textiles[18] and metal-semiconductor-metal metasurfaces[17,19]. However, these metamaterial designs have predominantly relied on conventional empirical approaches or manual optimization techniques[19–23]. In addition, the need for flexible color and infrared signal adaptivity across various background scenarios, such as soil, desert, and green vegetation, has not been adequately addressed or demonstrated to date.

Here, we present a general material-informatics-based[24,25] (MI) framework to inversely design the Ge/ZnS multilayer metamaterials for achieving the troublesome sophisticated reflectivity/emissivity spectra for VIS-IR camouflage. Our inverse design protocol integrates the Bayesian optimization[26] (BO) algorithm with the transfer matrix method[27,28] (TMM), enabling the automatic refinement of Ge/ZnS multilayer metamaterial structures through machine learning. To validate the predictions made by the TMM, we experimentally fabricate a Ge-ZnS multilayer metamaterial on a quartz substrate and characterized its visible reflectance and infrared emissivity. The results confirm the accuracy of the TMM predictions and are further supported by electromagnetic field simulations. The material informatics framework we present here can be extended to the design of other complex metamaterials and metasurfaces, enabling precise control over multispectral optical properties.

## Results

### Roadmap of Metamaterial Design

The roadmap of the material-informatics-based design for multilayer metamaterials is shown in Fig. 1a, which comprises encoding multilayer metamaterials into binary digits, physical simulation with the TMM, both visible and infrared camouflage evaluations, sensitivity of emissivity for thickness evaluation, figure-of-merit (FOM) characterization, and BO. The metamaterial structure comprises aperiodic Ge and ZnS multilayers grown on a quartz substrate and a topmost ZnS layer. The topmost ZnS layer is used for visible light camouflage, whereas the underlying Ge/ZnS multilayer was used for infrared camouflage. The thickness of the topmost ZnS layer ($d_{top}$) is critical for color displays since the changes in the thickness of ZnS can affect the resonance conditions of visible light at different wavelengths, leading to diverse reflectivity spectra with a tunable color appearance. Therefore, we can change $d_{top}$ to tune its color to adapt to different background colors. The thickness of the topmost ZnS layer is optimized to match that of the target color to achieve diverse color appearances. As for the aperiodic alternating Ge/ZnS multilayers, Ge and ZnS are chosen because of their relatively large difference in the infrared refractive index, which can be used to tune the emissivity spectra in the target infrared waveband effectively. For

simplification, the thickness of each Ge and ZnS sublayer is fixed at 200 nm, whose optical path is approximately one-quarter of the target wavelength in the atmospheric window, to induce low emissivity. In our optimization, the maximum number of sublayers is set to 10, and the thickness range of the topmost ZnS sublayer is set from 20 to 600 nm with a spacing of 10 nm, owing to accurate manufacturing considerations. Therefore, the total number of possible candidate multilayer metamaterials is $2^{10} \times 59 = 60416$. More details on how such a structural configuration is set up can be found in Supplementary Note 1. The dielectric function data are referred to in the optical handbook[29,30] and provided in Supplementary Fig. 1. With these material and structural configurations, the MI-based inverse design process is implemented as follows.

Firstly, we encode each Ge sublayer as a digit of 0 and each ZnS sublayer as a digit of 1. Each multilayer metamaterial can be denoted as a series of binary digits with the thickness of topmost ZnS layer. The reflectance in the visible band and emittance spectra in the infrared band of each metamaterial structure can be simulated using the transfer matrix method (TMM)[27,28], which is an efficient method for solving Maxwell's electromagnetic equations under layered structure conditions.

Secondly, we evaluate the designed metamaterials based on visible and infrared camouflage performances, and thickness sensitivity. The ideal visible camouflage requires designing a structure with a color similar to the ambient environment. Therefore, a target ambient color should be determined in advance; for example, a blackish-green color in the wild. The color display of the metamaterial structure is generated by mixing the reflected light in the visible band (0.38–0.78 μm). After we obtain the reflectance spectra in the visible band, the color is evaluated by calculating the tristimulus values ($X, Y, Z$) in the Commission Internationale de l'Elcairage (CIE) color space. The tristimulus values are calculated by integrating the spectral power distribution of the standard D65 illumination, reflectance spectra of the designed metamaterials, and the color-matching function. To measure the color difference, we transforme the tristimulus values in the CIE-XYZ color space into the lightness, chroma, and hue in the CIE-LCH color space. The color difference between the target color and the obtained color is then evaluated by

$$\Delta E_{CMC(l:c)} = \sqrt{\left(\frac{L_1 - L_2}{l \cdot S_L}\right)^2 + \left(\frac{C_1 - C_2}{c \cdot S_C}\right)^2 + \left(\frac{H_1 - H_2}{S_H}\right)^2} \quad (1)$$

where ($L, C, H$) are the color coordinates in the CIE-LCH color space, and Subscripts 1 and 2 denote the target and designed colors, respectively. Coefficients $l$ and $c$ are two weight factors used to amend the lightness and chroma differences to approximate the human sensation of color difference. Note that the smaller the $\Delta E_{CMC(l:c)}$, the smaller the color difference between the target and obtained colors. Further details of the color characterization and color difference calculation process are presented in Supplementary Note 2.

Since the main purpose of infrared camouflage is to reduce the thermal radiation power in the atmospheric window to be as close as possible to that of the ambient environment, infrared camouflage is highly dependent on the temperature of the object and environment, especially according to the Stefan–Boltzmann law, which states that the thermal radiation power is proportional to the fourth power of temperature. The typical infrared camouflage evaluation that only considers the selective emissivity spectrum is not thoughtful and practical enough; hence we evaluate the infrared camouflage by considering the emissivity spectra and object temperature simultaneously. Suppose the designed metamaterial and blackbody are in the same environment. Then we define a dimensionless infrared camouflage evaluation factor, ratio of thermal signal (RTS), to denote the

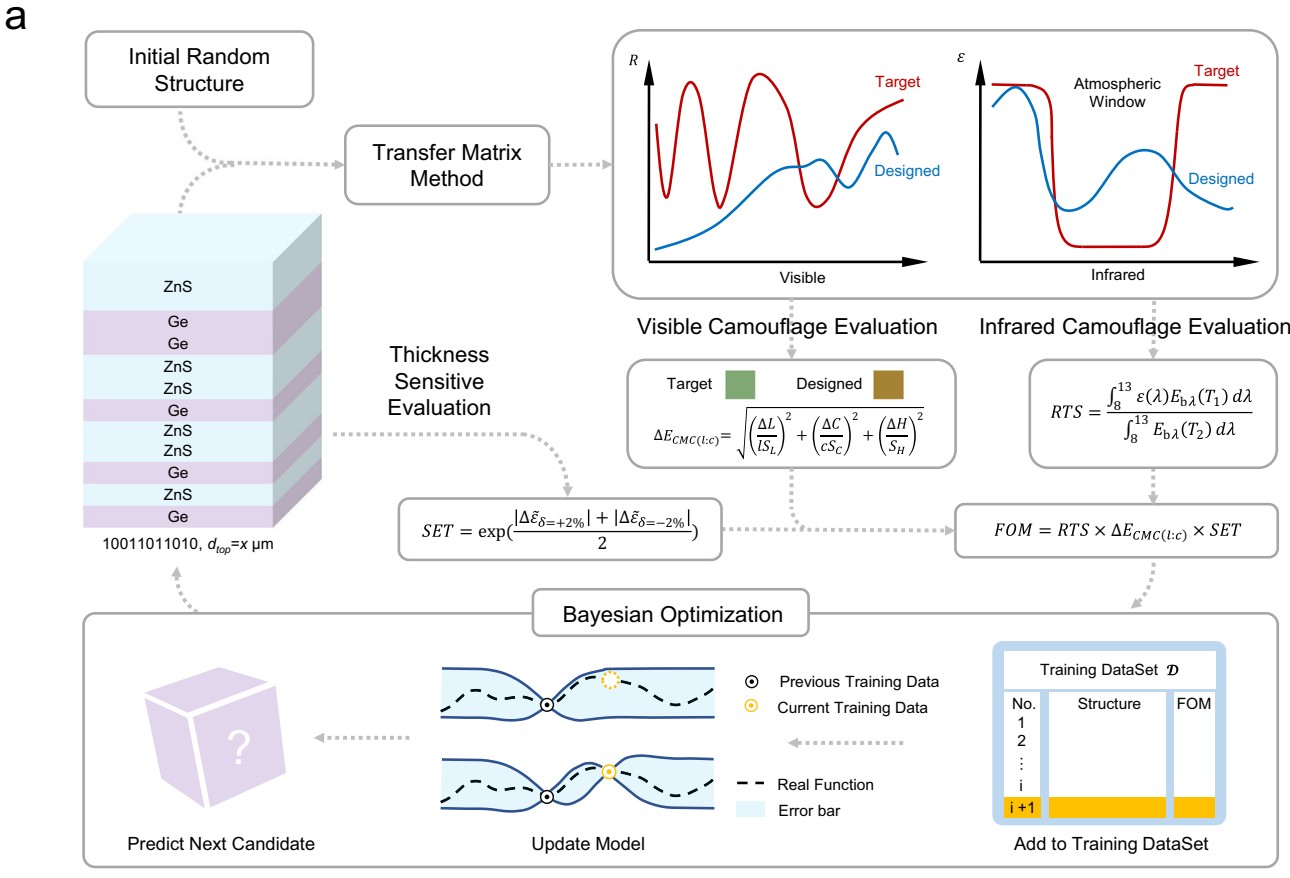

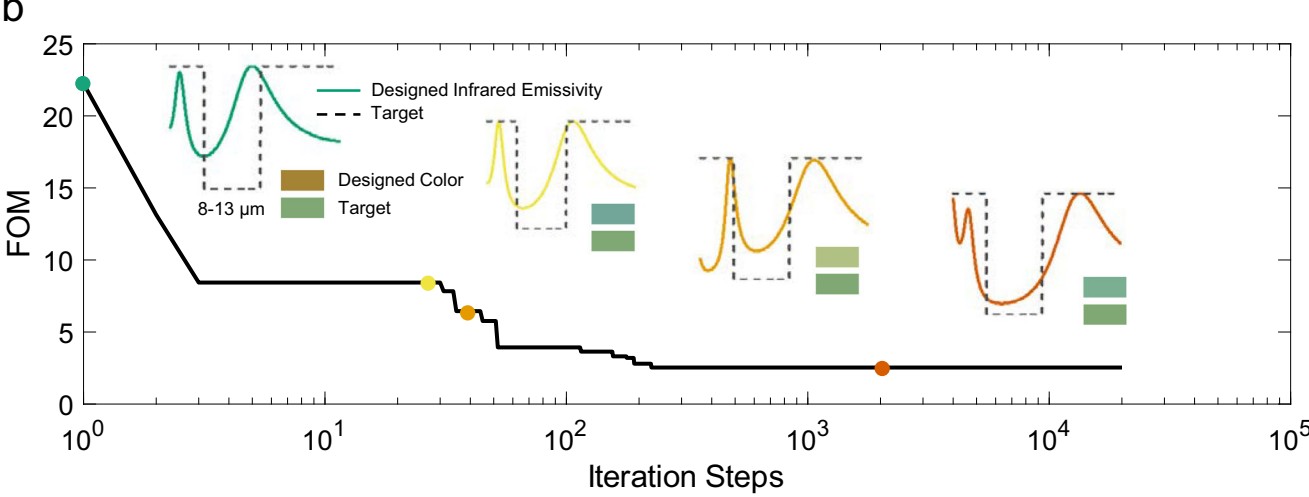

**Fig. 1 | Roadmap of inverse design of metamaterials for VIS-IR camouflage using material informatics. a** Schematic of the roadmap, where $d_{top}$ represents the thickness of the top layer. The meaning of other abbreviations can be found in the explanation following Eqs. 1, 2, and 7. **b** Evolution of the FOM with each iteration step. Inset: Color display and emissivity spectrum of four intermediate structures during the optimization process.

ratio of the thermal radiation power of the designed metamaterial to the blackbody in the same surroundings (but not the same temperature) as

$$\text{RTS} = \frac{\int_8^{13} \varepsilon(\lambda)\, E_{b\lambda}(T_1)\, d\lambda}{\int_8^{13} E_{b\lambda}(T_2)\, d\lambda} \qquad (2)$$

where $E_{b\lambda}$ is the monochromatic radiation of the blackbody and $\varepsilon(\lambda)$ is the monochromatic emissivity of the metamaterials calculated by the TMM algorithm. $T_1$ and $T_2$ represent the temperatures of the metamaterials and blackbody in the same environment, respectively, quantified by the thermal equilibrium of the metamaterials/blackbody and the ambient environment. At thermal equilibrium, the inner heating power $P_{inner}$, inward radiation power from the ambient $P_{amb}$,

outward heat convection power $P_{conv}$, and outward radiation power $P_{rad}$ are balanced $P_{inner} + P_{amb} = P_{conv} + P_{rad}$, and specifically

$$P_{amb}(T_{amb}) = \int d\Omega \int_0^\infty I_{BB}(T_{amb}, \lambda) \alpha(\lambda, \theta) \varepsilon_{atm}(\lambda, \theta) \cos(\theta) \, d\lambda \quad (3)$$

$$P_{conv}(T, T_{amb}) = h(T - T_{amb}) \quad (4)$$

$$P_{rad}(T) = \int d\Omega \int_0^\infty I_{BB}(T, \lambda) \varepsilon(\lambda, \theta) \cos\theta \, d\lambda \quad (5)$$

where $\int d\Omega = \int_0^{\pi/2} d\theta \sin\theta \int_0^{2\pi} d\phi$ is the angular integral of the hemisphere. $I_{BB}(T, \lambda)$ is the spectral radiance of a blackbody at temperature $T$. $\varepsilon(\lambda, \theta)$ and $\alpha(\lambda, \theta)$ are the emissivity and absorptivity spectra of the designed metamaterials which obey the Kirchhoff's law. $\varepsilon_{atm}(\lambda, \theta) = 1 - \tau(\lambda)^{1/\cos\theta}$ is the spectral emissivity of the atmosphere and $\tau(\lambda)$ is the atmospheric transmittance in the zenith direction[31]. $h$ is the convective heat transfer coefficient. By solving the thermal equilibrium equation, we can obtain the equilibrium temperature of the designed metamaterials ($T_1$) and blackbody ($T_2$) and then evaluate the corresponding infrared camouflage performance RTS according to Eq. 2. Compared to the other infrared camouflage evaluation factor, which only requires low emissivity in 8–13 μm and high emissivity in 5–8 μm and 13–20 μm, the proposed RTS here takes object temperature into consideration; hence RTS is superior to the selective emissivity for more accurate and practical infrared camouflage characterization. In a varying environment where the convective heat transfer coefficient or ambient temperature does not remain constant, the transient lumped model of the real-time temperature of the object should be considered rather than the thermal equilibrium steady-state temperature, as follows:

$$\rho c V \frac{\partial T}{\partial \tau} = P_{inner} + P_{amb} - P_{conv} - P_{rad} \quad (6)$$

where $\rho$, $c$, and $V$ are the density, specific heat, and volume of the object, respectively. The real-time temperature was averaged and substituted into Eq. 2 to obtain the RTS and evaluate the infrared camouflage performance.

In addition to the visible and infrared camouflage performances, the thickness sensitivity of the experimental fabrication is evaluated and fed back to the MI optimization for the first time. Because of fabrication errors, the measured emissivity of the fabricated samples always deviates from the theoretical designs to some extent. In other words, the structure's emissivity spectrum changes when the thickness of each layer is altered, which is evaluated by the sensitivity of emissivity to thickness (SET). A smaller SET indicates lower fabrication error and better matching between the measured and predicted emittance. Therefore, we consider SET for MI optimization, which can enhance the tolerance of design structures to fabrication errors. The SET of the layered structure is designed as follows:

$$SET = \exp\left(\frac{|\Delta\bar{\varepsilon}_{\delta = +2\%}| + |\Delta\bar{\varepsilon}_{\delta = -2\%}|}{2}\right) \quad (7)$$

where $\Delta\bar{\varepsilon}_{\delta = \pm 2\%}$ is the difference between the average emissivity of the design structure and the counterpart of the structure with a ± 2% thickness error for each layer based on the design structure.

For a better VIS-IR camouflage performance, $\Delta E_{CMC(l:c)}$ and RTS should be as small as possible. The SET should also be as small as possible to enhance the tolerance of the design structures to fabrication errors. Therefore, we define the FOM as the product of $E_{CMC(l:c)}$, RTS, and SET. The product of the first two terms indicates equal importance of visible and infrared camouflage, while the SET, which

ranges from 1 to $e$, can be regarded as a penalty factor to deprecate structures with low tolerance to thickness error. Consequently, a smaller FOM indicates a better VIS-IR camouflage performance and better consistency between the simulation and experiment, which is preferred and ready for optimization.

Thirdly, we employ BO[26] to accelerate the inverse search for the structure with the best VIS-IR camouflage performance. The BO can automatically select inputs that will yield better outputs through the trained Gaussian process, with the highest probability of obtaining better performance from the listed candidate digit series inputs under the machine-learning kernel. The listed candidate digit series inputs in our study are denoted as $\{X_i\}_{i=1,2,...N}$, where $X_i$ is a $d$-dimensional vector (11 here, including the topmost ZnS thickness) and $N$ is the total number of all metamaterial candidates (60,416 here). The output value is expressed as $FOM_i$ when $X_i$ is the input. As initial training data, $M$ (= 50 here) randomly chosen candidates are calculated, and their output FOMs are collected as the training data $D = \{X_k, FOM_k\}_{k=1,2,...,M}$. BO selectes the next $M+1$-th data $X_{M+1}$ with the help of machine learning, and the corresponding next $M+1$-th pair ($X_{M+1}$, $FOM_{M+1}$) is added to the training data $D$. To accelerate the BO process, random feature map[32] is employed, which allows us to approximate the Gaussian kernel function with a positive definite symmetric function by probabilistic sampling, and the two hyperparameters in the Gaussian process are automatically determined by maximizing the Type II likelihood[33]. Moreover, the Cholesky decomposition[34] and Thompson sampling[35] are employed to simplify the matrix operation and shorten the computation time.

The metamaterial structure could be optimized efficiently by minimizing the FOM through the BO process. The evolutional minimal FOM versus iteration step are plotted in Fig. 1b. During the BO process, the corresponding FOM is rapidly decreased to achieve the optimal value within iteration steps <1% of the total candidates. No better result other than this optimal result in our design space is found through ~30% of the total candidates simulated. Along the evolution of FOM, four intermediate metamaterials are also shown in Fig. 1b with their color display and infrared emissivity spectra. Along the FOM evolution, the designed color is getting closer to the target one, and the corresponding infrared emissivity spectrum is also approaching the ideal emissivity spectrum for infrared camouflage with decreasing RTS. Therefore, the metamaterial for VIS-IR camouflage can be designed efficiently under the material informatics inverse design roadmap.

To further demonstrate the merits of the proposed FOM, we performed another round of metamaterial optimization with $FOM_2 = RTS \times SET$, aiming at a better infrared camouflage and better consistency between the simulation and experiment without visible camouflage, as shown in Fig. 2a. Accordingly, the structure is only a 10-layer Ge/ZnS multilayer metamaterial, excluding the colorful topmost ZnS layer. Using the same BO process, the RTS evolution in the first and second rounds of optimization is shown differently in Fig. 2b. Taking RTS × SET as $FOM_2$, the RTS in the second round decreases with the iteration steps. In contrast, $FOM_1$ in the first round is the product of RTS, $\Delta E_{CMC(l:c)}$, and SET, thus the RTS fluctuates with the iteration steps, as shown in Fig. 1b. Moreover, the RTS in the second round is smaller than that in the first round, which can be easily attributed to the tradeoff between the color difference $\Delta E_{CMC(l:c)}$ and the infrared camouflage performance RTS. Through two rounds of optimization, two metamaterial structures, 800 nm Ge/400 nm ZnS/800 nm Ge (Struct. 2) and 340 nm ZnS/600 nm Ge/800 nm ZnS/600 nm Ge (Struct. 1) are obtained. The visible reflectivity spectra and corresponding colors are plotted in Fig. 2c. By taking RTS × SET as the $FOM_2$, the designed metamaterial has a gray color, which can only be used for infrared camouflage, while by taking RTS × $\Delta E_{CMC(l:c)}$ × SET as the $FOM_1$, the resultant green-colored metamaterial can be applied for VIS-IR camouflage. This comparison authenticates the necessity and superiority of the proposed multi-objective optimization.

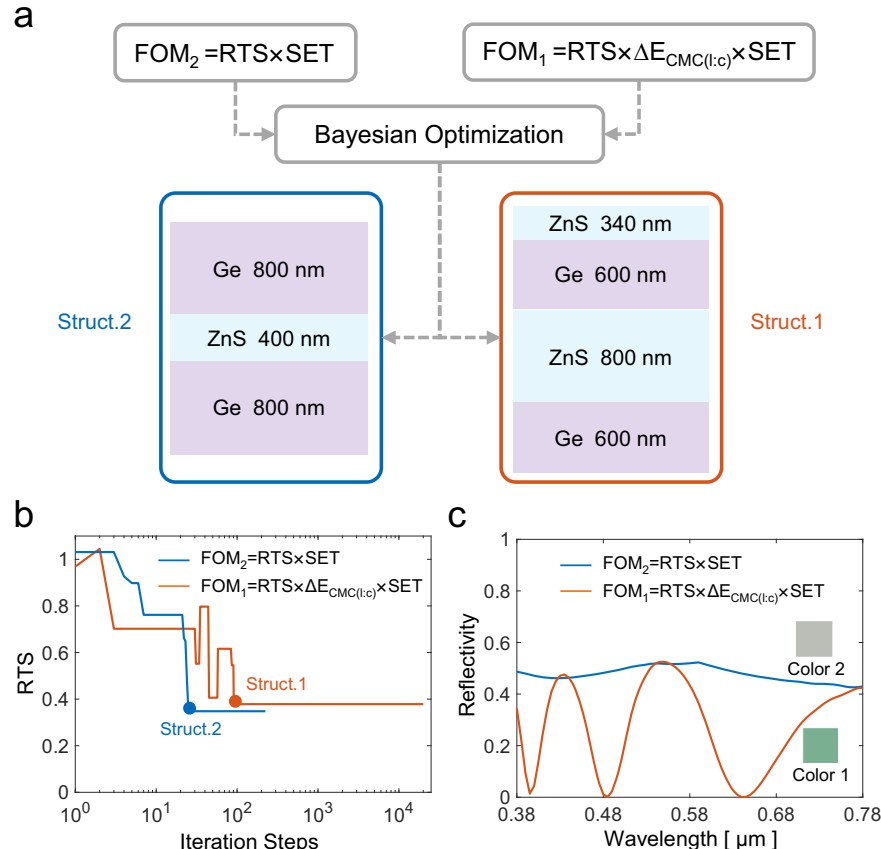

**Fig. 2 | Comparison between optimization with $FOM_2 = RTS \times SET$ and optimization with $FOM_1 = RTS \times \Delta E_{CMC(l:c)} \times SET$. a** Schematics illustrating the two metamaterial structures. **b** Comparison of the evolution of RTS during the iteration process. **c** Comparison of the visible reflectivity spectra and corresponding colors, depicted in the inset. The definition of RTS, $\Delta E_{CMC(l:c)}$, and SET can be found in Eqs. 1, 2, and 7, respectively.

It should be noted that if more topmost layers are introduced for more color and diverse visible camouflage effects, the camouflage requirements for visible and infrared can be optimized in a decoupled manner. However, this only works in a layered structure because it can be divided into two parts related to visible and infrared camouflage. Qi et al. empirically designed a Ge/ZnS layered structure to achieve infrared camouflage with color[36]. The topmost ZnS layer corresponds to the color display, whereas the rest corresponds to infrared camouflage. Sheng et al. designed a top $SiO_2/SiN$ emitter for infrared emissivity tuning and a bottom $MgF_2/SiC$ Tamm structure for visible emissivity tuning[37]. However, for other structures (such as gratings), it is unclear which part of the structure corresponds to visible or infrared emissivity, thus deactivating the decoupling design. Therefore, coupling optimization helps to clarify the corresponding relationship between the structure and spectrum and aids in saving extra work by integrating the two parts of the results obtained by decoupled optimization. Here, the coupling optimization scheme is adopted considering the universality and extensibility of the inverse design algorithm such that it can be applied to broader application scenarios, such as designing VIS-IR-laser-radar-compatible camouflage with gratings, rods, square patterns, and so on.

**Visible camouflage performance**

Using the above roadmap for VIS-IR camouflage, we execute 60 rounds of optimization with different colors, whose color coordinates in the CIE 1931 color space are shown in Fig. 3a. The purple dashed line is the envelope curve encircling all simulated structures, which indicates the feasible color range that our structure can achieve. Because there is only one ZnS layer for visible camouflage, the achievable color is

limited and is enclosed by the purple dashed line. A high-chroma color outside the achievable range, such as scarlet red, is difficult to achieve because of the limitations of a single ZnS layer. Nevertheless, conventional camouflage color patterns in typical surroundings, such as forests (olive drab) and deserts (earth yellow), can be easily obtained with our present structures, which proves the broad application prospects of our metamaterial structures for VIS-IR camouflage. The color range can be expanded by integrating more complicated structure patterns and materials in our inverse design framework. A relevant discussion is provided in Supplementary Note 3. Furthermore, eight metamaterial structures are randomly selected for fabrication and experimental validation, with their corresponding target color, designed color, simulation error (the color difference between the target and design), and visible reflectivity spectra plotted in Fig. 3b. The measured visible reflectivity data are provided in the Supplementary Data. The spectra prove that the designed structure displays color via selective reflectance in the visible region. The similarity of the designed color to the target color and sufficiently small color difference indicates the accuracy of our MI-based algorithm and excellent visible camouflage performance. The coordinates of the designed colors are also marked in Fig. 3a with black stars.

Moreover, we precisely deposit ZnS and Ge layer by layer using electron-beam evaporation to manufacture the selected samples, and their thicknesses are calibrated using spectroscopic ellipsometry. An SEM image of the No. 8 fabricated sample is shown in Fig. 4c as a typical example to verify its manufacturing accuracy. A four-layer structure is observed, and the thickness error of each layer is within an acceptable range. Figure 3c shows the color of the samples under natural light with an experimental error (the color difference between the target

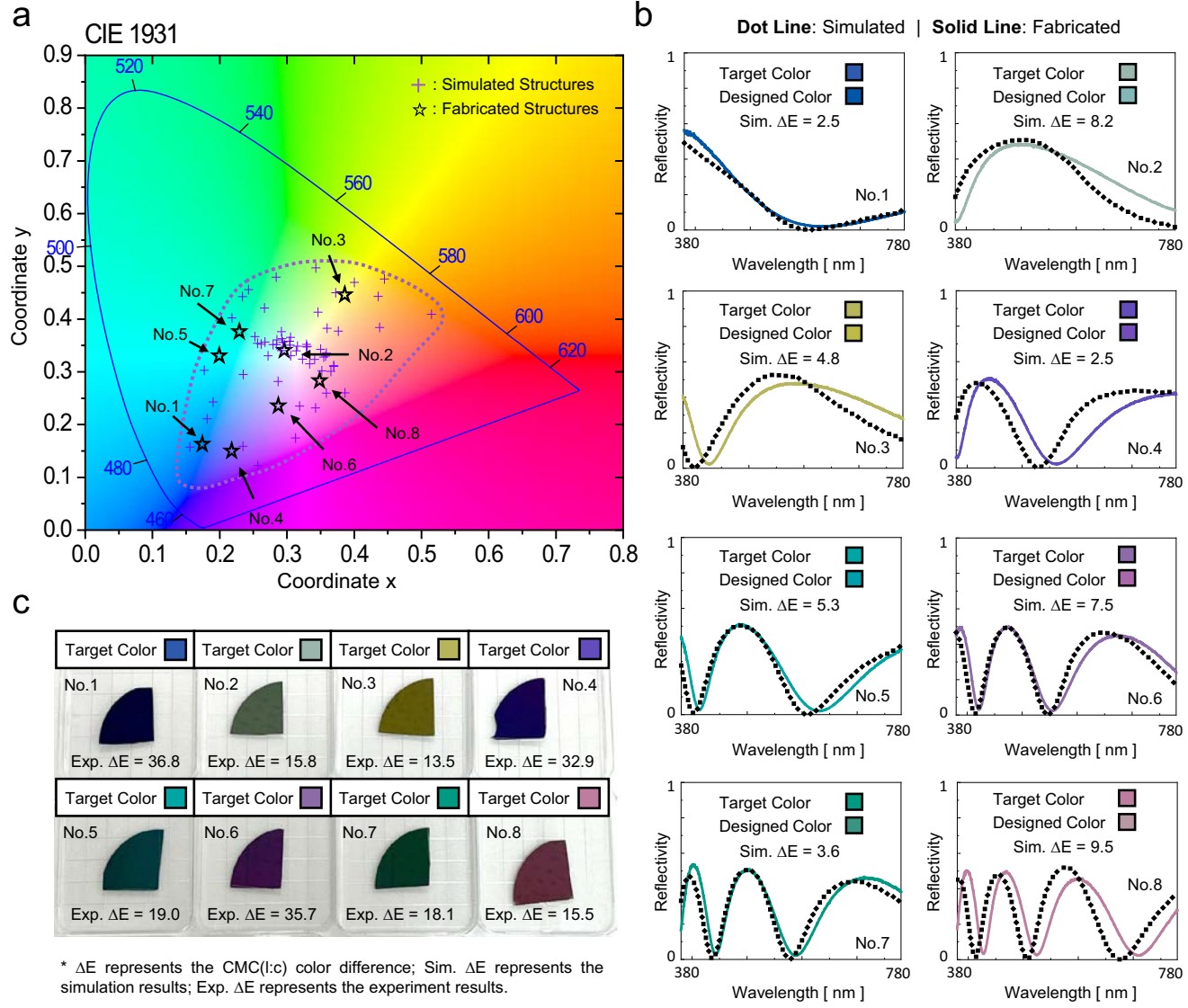

**Fig. 3 | Color matching performance of both designed structures and fabricated samples. a** Color range and distribution of designed structures in CIE 1931 color space. **b** Simulated and measured reflectivity spectra of eight structures within the visible waveband, along with their target color, designed color, and the corresponding CMC(l:c) color difference, denoted as Sim. ΔE. **c** Photographs showcasing the fabricated structures alongside their target color, fabricated color, and the respective CMC(l:c) color difference, denoted as Exp. ΔE.

color and the fabricated color), which is caused by both the fabricated thickness deviation and different lighting conditions. Through experimental verification, our MI-based inverse algorithm is proven effective in designing metamaterials that satisfy visible camouflage with a wide range of colors.

**Infrared camouflage performance**

The infrared camouflage performances of eight selected metamaterials are simulated and experimentally demonstrated. Figure 4a shows the simulated and measured emissivity spectra of the eight selected metamaterials in the infrared band, the agreement of which implies the effectiveness of our MI-based inverse design algorithm in designing infrared camouflage. The measured infrared emissivity data are presented in the Supplementary Data. All metamaterials exhibit low emissivity below 0.2 in the atmospheric window (8–13 μm) and high emissivity outside. The abrupt increase of emissivity at ~9 μm originated from the phonon–polariton resonance of the quartz substrate. Concave-shaped emissivity spectra are obtained by minimizing the RTS factor during optimization. The resulting emissivity spectrum squeezes the thermal radiation power in the atmosphere window by directly lowering the emissivity in 8–13 μm and enlarging the radiative heat dissipation to reduce the object temperature as much as possible. A comparison of the infrared camouflage performance of our optimized structure and an empirically designed structure for VIS-IR camouflage is presented in Supplementary Note 4, which demonstrates the superiority of our optimal design method. For further verification, the normalized electric field intensities of sample No. 8 at different resonant wavelengths are plotted in Fig. 4b. The intensity of the electric field decays heavily at 10 μm in the atmospheric window. However, the intensity is maintained in the visible (0.5 μm) and other non-atmospheric windows (5 μm and 15 μm). This is because the structure forms a forbidden band in 8–13 μm, resulting in low absorbance and emittance in the atmospheric window but relatively high absorbance and emittance outside with lossy quartz substrate.

IR experiments are performed to confirm the actual infrared camouflage performance of the fabricated samples. A schematic of the experimental setup is shown in Fig. 5a. Eight fabricated samples with different colors are placed on a heating plate with a bare quartz

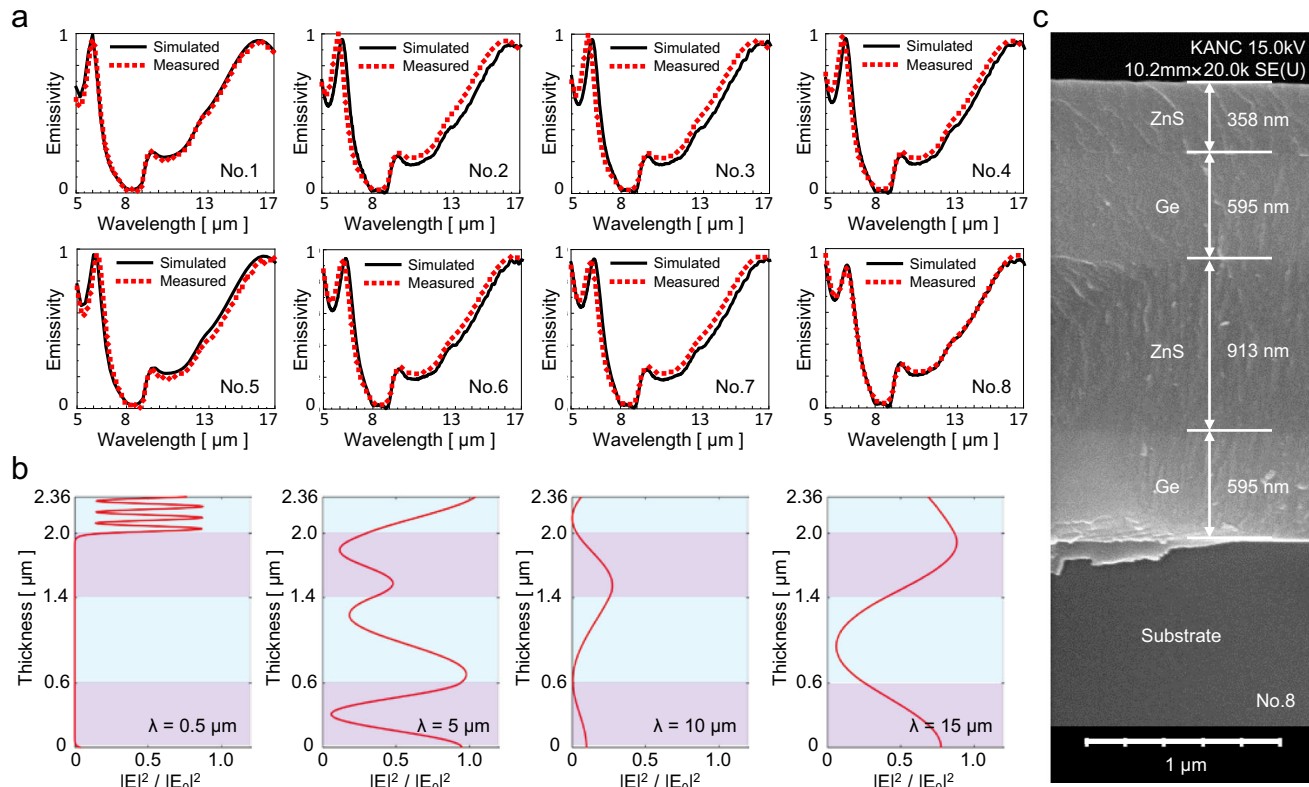

**Fig. 4 | Demonstration of infrared camouflage performance. a** Comparison of simulated and measured emissivity spectra in the infrared waveband for eight designed structures. **b** Profile of normalized electric field intensity for Sample No. 8 at various resonant wavelengths ($\lambda = 0.5, 5, 10,$ and $15\,\mu m$). **c** SEM image of Sample No. 8, where the thickness of each layer is indicated.

substrate and a black carbon tape/quartz substrate for comparison. Carbon tape and bare quartz are used as comparative objects without the infrared camouflage function, which display normal thermal signals at the same temperature. All the experimental samples achieve an approximate temperature under thermal equilibrium with the heating plate; however, the camouflaged metamaterials demonstrate lower thermal signals owing to their infrared camouflage function. The angle-insensitive properties are verified by acquiring infrared images at different observation angles. Figure 5b shows infrared images of the heating experiments. The samples and observation angles are marked on the header, and the temperatures of the heating plate are marked on the left side. All our fabricated samples exhibit lower thermal signals than bare quartz and carbon tape/quartz at each observation angle, achieving infrared camouflage performance regardless of the color and observation angle. As the observation angle varies, the temperatures of our metamaterial samples are still lower, indicating an angle-independent and stable infrared camouflage performance. This angle-independent property is further demonstrated by the emissivity in Fig. 5c. The measured infrared emissivity spectra of sample No. 8 under different incident angles are plotted, which are close to each other, implying good angle-independent infrared camouflage performance. Data on the measured infrared emissivity at different incident angles are shown in the Supplementary Data. Simulation results that further prove the angle-independent properties are shown in Supplementary Note 5.

Overall, the emissivity spectrum and infrared experiments demonstrate our designed metamaterials' good angle-independent VIS-IR camouflage performance. Our study develops an efficient machine-learning-kernel inverse design framework with strong versatility and consider nanoscale fabrication errors for VIS-IR-camouflaged metamaterials. Compared to the representative work in the field of

camouflage[38], our work advances with more colors, a simpler structure, and better VIS-IR camouflage performance. A more detailed discussion is provided in Supplementary Note 6.

In summary, we present an efficient material informatics inverse design framework to design Ge/ZnS multilayer metamaterials for visible-infrared-compatible camouflage, not only achieving stringent adjustment of visible reflectivity spectra and infrared emissivity spectra simultaneously for VIS-IR camouflage but also obtaining broad color and infrared signals with background-matching characteristics. In the implementation of the material informatics inverse design, we encode the multilayer metamaterial as binary digits, characterize the visible and infrared spectra, evaluate the VIS-IR camouflage and sensitivity of emissivity to thickness, and employ machine-learning-kernel Bayesian optimization. The optimal metamaterial structure in our design space can be obtained with iteration steps of <1% of the total candidate structures. Eight different-colored metamaterials are designed, fabricated, and measured to validate the angle- and temperature-independent VIS-IR performance. The present VIS-IR metamaterials exhibit promising potential in multispectral camouflage applications, and more importantly, the present ultrahigh-efficient material informatics inverse design framework can be extended to other multi-objective optimization problems beyond multispectral camouflage.

## Methods
### Simulations
Our structures' visible reflectivity, infrared emissivity, and normalized electric-field intensity profiles were simulated using the TMM. Color simulations were performed using MATLAB with our code, the principle of which can be found in Supplementary Note 2 in the SI. BO was performed using Python with the help of the open-source package PHYSBO.

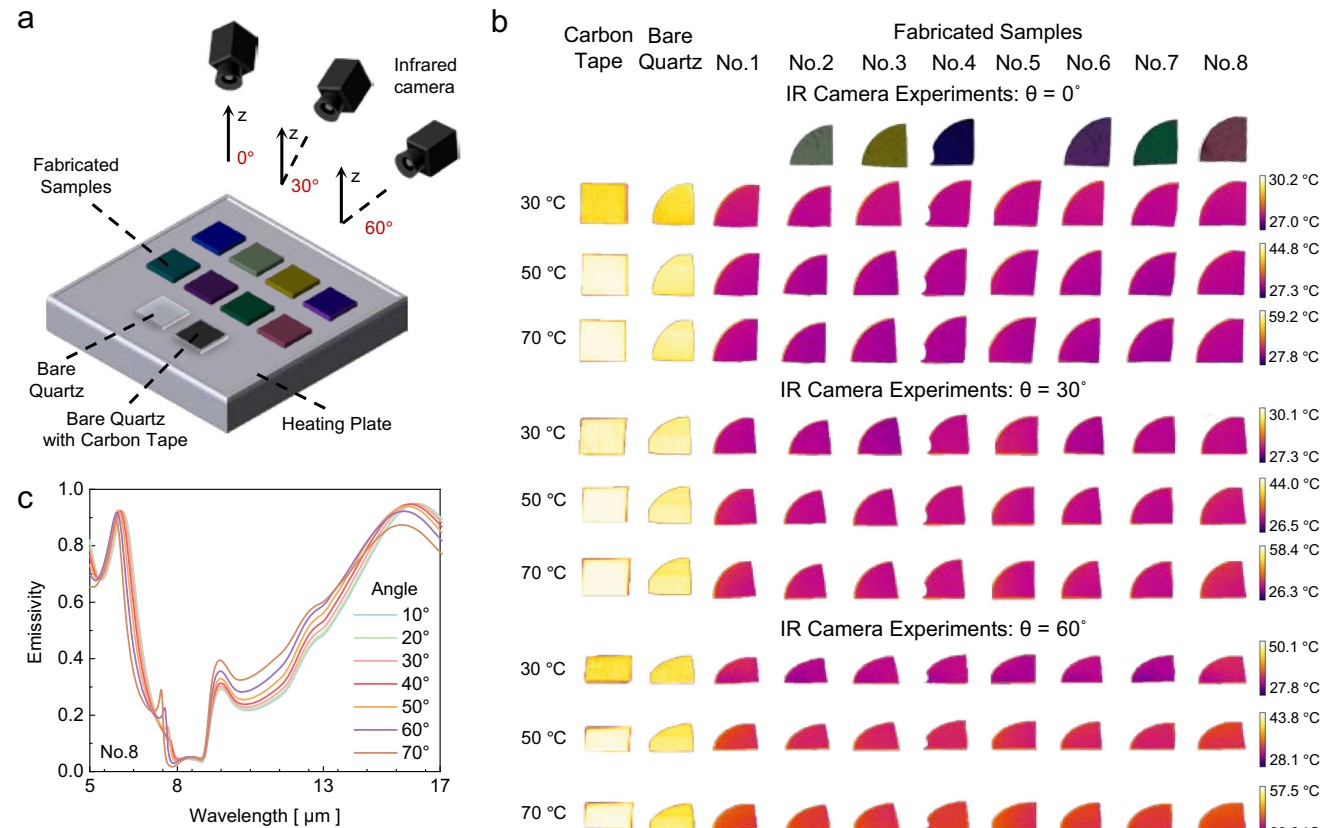

**Fig. 5 | Infrared experiment and angle-independent infrared camouflage performance. a** Schematic of the experimental setup. **b** Infrared images captured at different temperatures and observation angles using a longwave (7.5–13.5 μm) infrared camera (FLIR A655SC), showing the infrared response of eight fabricated samples. All samples underwent simultaneous infrared thermography imaging under identical temperature conditions. The infrared thermography images were normalized using the maximum intensity observed for the carbon tape-coated quartz at θ = 0°. **c** Measured infrared emissivity spectra of Sample No. 8 for varying incident angles.

## Samples fabrication

The Ge/ZnS multilayer samples were fabricated on p-doped double-side-polished <100> crystalline silicon substrates (Hi-Solar Co., LTD.). The fabrication process involved the alternate deposition of Ge and ZnS layers with predetermined thicknesses on the Si substrates using an electron beam evaporator (UEE, ULTEC). During deposition, the deposition rate and accumulated thickness of each layer were continuously monitored using a quartz crystal monitor. The film thicknesses were subsequently validated using a spectroscopic ellipsometer (Elli-SE, Ellipso Technology), ensuring accuracy and precision in the measurements.

## Optical characterization

The visible reflectivities of the Ge/ZnS multilayer samples are measured using a spectrophotometer (Cary5000, Thermo Fisher). The infrared emissivity of the Ge-ZnS multilayer samples was measured using a Fourier transform infrared spectrometer (INVENIO R, Bruker) equipped with a variable-angle accessory (Harrick Seagull).

## Infrared experiment

Figure 5b presents longwave (7.5–13.5 μm) infrared thermography images, which were acquired utilizing an infrared camera (FLIR A655SC, Teledyne FLIR). The temperatures of the fabricated eight multilayer samples, along with two reference samples (carbon tape-coated quartz and bare quartz), were precisely maintained at 30, 50, and 70 °C. All samples were subjected to simultaneous infrared thermography imaging at the same temperature settings. To explore the angle-dependent thermal emission characteristics, the infrared camera was systematically rotated from 0° to 60° in 30° increments. For expedient data acquisition, image analysis, and data extraction, a thermal analysis software (ResearchIR, Teledyne FLIR) was employed. The infrared thermography images were normalized based on the maximum intensity observed for the carbon tape-coated quartz at θ = 0°. Meanwhile, the sample temperatures were ascertained through the use of adhesive thermocouples (SA1-K-72, Omega Engineering).

## Data availability

The measured reflectivity data in the visible band and emissivity in the infrared band are provided in the Supplementary Dataset 1 file.

## Code availability

The corresponding code for BO is available at https://github.com/issp-center-dev/PHYSBO, distributed under the GNU General Public License version 3 (GPL v3).

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

## Acknowledgements

We acknowledge financial support from the National Natural Science Foundation of China (52211540005, 52076087, 52161160332), the Open Project Program of Wuhan National Laboratory for Optoelectronics (2021WNLOKF004), the Science and Technology Program of Hubei Province (2021BLB176), and the Knowledge Innovation Shuguang Program. S.-K.K. acknowledges financial support from the National Research Foundation of Korea through the Basic Science Research Program (RS-2023-00207966) and the Nano-Material Technology Development Program (2022M3H4A1A02046445). J.S. acknowledges financial support from JSPS Bilateral Joint Project (120227404).

## Author contributions

W.X. and R.H. conceived the study. W.X., S.Y., Z.C., and R.H. designed the roadmap, performed numerical simulations, and developed the code. Y.-J.L. and S.-K.K. designed and performed the experiments. W.X. and R.H. wrote the manuscript and analyzed the numerical and experimental results. S.-K.K. and R.H. supervised the study. J.S., S.-K.K., and R.H. revised the manuscript. All the authors contributed to the discussion and revision of the manuscript.

## Competing interests

The authors declare no competing interests.
