## [Peer Review File · Nature Communications]

Ultrahigh-efficient material informatics inverse design of thermal metamaterials for visible-infrared-compatible camouflageReviewer #1 (Remarks to the Author):

In this manuscript, a visible-infrared-compatible camouflage thermal metamaterial is designed using a high-efficient material-informatics inverse design framework. The designed structure can not only display wide range of visible colors, but also exhibit satisfying thermal camouflage effect. The model construction, electromagnetic simulation, optimization process, samples fabrication, and VIS-IR camouflage performance validation were presented comprehensively. This paper is novel in the design method, effective in optimization, valid in the performance, and fluent in the writing. I suggest to accepting it after a minor revision.

- 1) In the first paragraph in 'Visible Camouflage Performance' section, the author claim that the color is limited due to the one-layer ZnS. Can the author supplement some discussion on how to expand the color range?
- 2) It would be better if the author can supplement a SEM image of the experimental sample for better verification with the designed structure.
- 3) Can the authors explain why the binary digits of 0 and 1 are used instead of the continuously tunable thickness of each sublayer? As a result, the searching space will become smaller and the optimal structure may be missed.
- 4) The authors defined a dimensionless IR camouflage evaluation factor RTS, which depends on the thermal equilibrium of the metamaterials and the ambient environment. Does it also work in an unknown or varying environment (e.g., the convective heat transfer coefficient, the ambient temperature)?
- 5) Compared to the existing works (e.g., ACS Appl. Nano Mater. 2022, 5, 5119–5127; ACS Appl. Mater. Interfaces 2022, 14, 24690–24696) on visible-to-infrared camouflage, how much performance improvement can be achieved by the inverse design protocol than conventional empirical design strategies?

6) There are some details that should be considered in a revision.

In Fig 1b and Fig 2b, it is better to change the x label to "Iteration steps" to make it more precise. In Fig 5b, the spacing of each row should be adjusted; the font size of Fig 5c is relatively too small and should be adjusted too.

The English writing should be better polished and some typos should be corrected.

7) Recently there are some new developments on thermal metamaterials. The authors are suggested to cite more references. To list a few:

Li, Y. et al. Transforming heat transfer with thermal metamaterials and devices. Nat. Rev. Mater. 6, 488-507 (2021).

Xu, G. et al. Diffusive topological transport in spatiotemporal thermal lattices. Nat. Phys. 18, 450-456 (2022).

Li, J. et al. Reciprocity of thermal diffusion in time-modulated systems. Nat. Commun. 13, 1-8 (2022).

Xu, L. et al. Thermal Willis Coupling in Spatiotemporal Diffusive Metamaterials. Phys. Rev. Lett. 129, 155901 (2022).

Reviewer #2 (Remarks to the Author):

In the manuscript entitled "Ultrahigh-efficient material informatics inverse design of thermal metamaterials for visible-infrared-compatible camouflage" by Wang Xi et al, a Bayesian optimization algorithm coupled with transfer matrix method is employed to design multilayer germanium (Ge) and zinc sulfide (ZnS) metamaterials for VIS-IR camouflage. They experimentally fabricated the Ge/ZnS multilayer metamaterial atop quartz substrate and measured the visible reflectance and IR emissivity. The experimental results agree well with numerical calculations. This

is a solid demonstration, but it lacks the novelty to be published in top journals like Nature Communications. In fact, Zhu et al have experimentally demonstrated multispectral camouflage for the visible, mid-infrared (MIR, 3–5 and 8–14 μm), lasers (1.55 and 10.6 μm) and microwave (8–12 GHz) bands with simultaneous efficient radiative cooling in the non-atmospheric window (5–8 μm) [Zhu, Huanzheng, et al. "Multispectral camouflage for infrared, visible, lasers and microwave with radiative cooling." Nature communications 12.1 (2021): 1-8]. Their device consists of a ZnS/Ge multilayer for wavelength selective emission and a Cu-ITO-Cu metasurface for microwave absorption. Their design is more sophisticated, more powerful compared to the design here in this manuscript where a similar ZnS/Ge multilayer was employed. At the same time, Bayesian optimization has also previously been employed for the design of metamaterials [see e.g, Sakurai, Atsushi, et al. "Ultrannarrow-band wavelength-selective thermal emission with aperiodic multilayered metamaterials designed by Bayesian optimization." ACS central science 5.2 (2019): 319-326]. As a result, I have to suggest the rejection of this manuscript.

Reviewer #3 (Remarks to the Author):

The paper titled "ultrahigh-efficient material informatics inverse design of thermal metamaterials for visible-infrared-compatible camouflage" uses multilayer Ge and ZnS metamaterials to achieve camouflage in visible and infrared bands at the same time. The authors harness Bayesian optimization to inverse design the distribution of Ge and ZnS in order to achieve targeted reflectance spectra for different colors. In my opinion, this work is interesting, complete, and potentially useful. However, I do have a few major comments from the material design and inverse design algorithm perspectives.

1. In this paper, the design space is defined as follows: a topmost ZnS sublayer with a thickness from 20 to 600 nm, and a total number of 10 sublayers which can be either ZnS or Ge. To me, the selection of such design space is very random and seemingly requires a lot of expert intuition. Can authors explain more about why we should focus on this particular design space? Two questions need to be addressed here: a) Is the design space expressive enough, i.e., can the metasurface cover all/most of the colors and IR? b) Is the design space concise enough, i.e., how many designs will result in meaningful metasurfaces within this design space? In my opinion, the selection of the "design space" is the most crucial part of this paper. With a well-chosen design space, most algorithms can easily perform the optimization task. More justification and physical intuition needs to be added for this point.

2. The authors mentioned in the introduction that "... these metamaterial designs are mainly dependent on conventional empirical design strategies or manual optimization, and thus their performances are far from that of the globally optimized one". However, there is no guarantee that the designs discovered by the authors are globally optimal. Such sub-optimal is two-fold: a) Since Bayesian optimization (BO) only evaluated 1% of the design space, it is unknown how good the design discovered by BO compared to the global optimal. b) the total 60416 designs only represent a small fraction of realizable designs. The optimal design among these 60416 designs is neither globally optimal. Therefore, I think the authors should be careful when using the word "optimal".

3. The authors compared two FOMs, i.e., FOM_1 and FOM_2, where FOM_2 = RTS. It seems that the VIS camouflage requirement does not affect IR camouflage performance much (only a few percent inferior). Is this true for all VIS camouflage colors? If this is true for all other colors, it may indicate that the camouflage requirements for VIS and IR might be decoupled in the current design space. From these observations, the authors might find some physical explanations for why such a design space works well.

4. I think the authors should also come up with some criteria characterizing the difference between targeted color and inverse-designed color. Such criteria should be used to measure both the simulation error and the experimental error. A photograph can be used to determine the color of an experimentally fabricated sample.

Minor comments:

Line 62, is the abbreviation "MIR" defined or not?

Response to Reviewers' comments

Responses to the comments of the First Reviewer:

Reviewer #1: In this manuscript, a visible-infrared-compatible camouflage thermal metamaterial is designed using a high-efficient material-informatics inverse design framework. The designed structure can not only display wide range of visible colors, but also exhibit satisfying thermal camouflage effect. The model construction, electromagnetic simulation, optimization process, samples fabrication, and VIS-IR camouflage performance validation were presented comprehensively. This paper is novel in the design method, effective in optimization, valid in the performance, and fluent in the writing. I suggest to accepting it after a minor revision.

Authors' Response:

Thanks very much for your positive comments.

1. *In the first paragraph in 'Visible Camouflage Performance' section, the author claim that the color is limited due to the one-layer ZnS. Can the author supplement some discussion on how to expand the color range?*

Authors' Response:

Thanks for the good suggestion. In our proposed structure, only a tunable ZnS layer is settled to demonstrate color for simplification. Therefore, the obtained color range is limited due to the relatively simple structure. In fact, there are many other structure patterns along with diverse materials which can be utilized to expand the color range. For example, we design a new topmost structure pattern to obtain color out of the current color range, whose structure diagram is shown in Supporting Figure R1(a). The topmost periodic structure, whose period is P , consists of a 70-nm-thick Al square pattern whose side length is L , a 35-nm-thick Ge layer and a 100-nm-thick Ag layer from top to bottom. Lower structure is one of our optimized structures. This new proposed structure pattern can be used to obtain colors which are not obtainable with the original layered structure pattern. For instance, we design three structures (α , β , γ) from the new proposed structure pattern and simulate their visible reflectance with RCWA (Rigorous Coupled Wave Analysis) to obtain their colors. The visible reflectance and colors are plotted in Supporting Figure R1(c) with the structure parameters labeled. The colors of three structures (α , β , γ) are marked as diamonds in CIE 1931 color space compared with the color range of the layered structures, which is lined out by the purple dotted line, as shown in Supporting Figure R1(b). It can be seen that the colors of our new proposed structures lie beyond the purple dotted line, which indicates an expansion of color range. It should be noted that the proposed structure here is just a demo. In fact, the 35-nm-thick Ge layer and the 100-nm-thick Ag layer which is fixed here for simplification can also be tunable. Moreover, grating, rods, rectangles, etc. can be designed as new structure patterns with more tunable parameters. It can be inferred that there are great chance to expand the color range after integrating such more complicated structure pattern in our inverse design framework. Necessary modification including simulation method and design space should be made to make whole framework works well. Also, the calculation

cost would be more expensive when more complicated structure is integrated. One should make tradeoff between the performance of the obtained structure and the feasibility of the whole optimization. Relevant discussion has been supplemented in Page 13, Paragraph 1, Line 13 in section ‘Visible Camouflage Performance’ and marked red. Details have been supplemented in Supplementary Note 4.

Supporting Figure R1 (also Supplementary Figure S2) (a) Schematic of the new proposed structure pattern. (b) CIE 1931 color space. Purple crosses: layered structures designed by our inverse design algorithm; Purple dotted line: the color range of the layered structures; Black diamonds: the structures with new proposed pattern. (c) Visible Reflectance of the three new proposed structures. Corresponding structure parameters and the colors are labelled.

2. *It would be better if the author can supplement a SEM image of the experimental sample for better verification with the designed structure.*

Authors’ Response:

Thanks for the good suggestion. We have added a SEM image of the No.8 fabricated sample as a representative in Figure 4(m), with the structure diagram shown in Figure 4(i). From the SEM image we can clearly observe the four-layer structure with each corresponding layer thickness matches well. We believe the supplementary SEM image can demonstrate our manufacturing accuracy and the consistency between our simulation and experiment. Relevant description has been supplemented in Page 13, Paragraph 1, Line 3 from bottom and marked red.

Supporting Figure R2 (also Figure 4) (a-h) The simulated and measured emissivity spectra in infrared waveband of eight designed structures. (i-l) Profile of normalized electric field intensity of the No.8 sample at different resonant wavelength (i) $\lambda=0.5 \mu\text{m}$, (j) $\lambda=5 \mu\text{m}$, (k) $\lambda=10 \mu\text{m}$, (l) $\lambda=15 \mu\text{m}$. (m) SEM image of the No.8 sample. Thickness of each layer is marked with the measuring scale.

3. Can the authors explain why the binary digits of 0 and 1 are used instead of the continuously tunable thickness of each sublayer? As a result, the searching space will become smaller and the optimal structure may be missed.

Authors' Response:

Thanks for the good question. The reason for using the binary digits of 0 and 1 instead of the continuously tunable thickness of each sublayer is to avoid making the searching space too large to solve.

We use binary digits of 0 and 1 in order to denote the different structures. Here we use the fixed layer thickness for simplification (although the search space is also as large as 2^{11}), but we also can use more other digits to consider more continuously tunable thickness of each sublayer. We agree with you a better structure may be found with the continuously tunable thickness of each sublayer. However, the larger searching space at the same time may be hard to solve. A searching space with continuously tunable thickness of each sublayer can be quantified as n^m (designing an m layers structure where each layer has n choices). Suppose a five-layer structure ($m=5$), and each layer thickness ranges from 20 nm to 1 μm with a spacing of 10 nm with two kinds of material choices ($n=99 \times 2$). Therefore, the design space is $(99 \times 2)^5 = 3.04 \times 10^{11}$, which requires 10^7 order of computing hour which is too large to solve. To make it solvable, a limited searching space is needed. The most effective way of reducing an exponential type searching space is to lower the base, which is the number of each sublayer's choices.

Therefore, we fix the thickness of each sublayer ($n=1 \times 2$) and loosen the number of sublayers to 10 ($m=10$), which can significantly lower the searching space to an order of 10^4 . Since each

sublayer only has two choices, the binary digits of 0 and 1 can be used to represent the structure. Although the limited searching space could not guarantee the optimal result, we can increase the probability of finding it by setting a reasonable value to the fixed sublayer thickness. In our research the value is set to be 200nm, which can ensure the opportunity to form Ge/ZnS pair, whose optical path is approximate to the quarter of target wavelength in the IR band to induce more constructive interferences.

In conclusion, we reduce the searching space to a solvable range by fixing the sublayer thickness and 01 sequence, and improve the probability of getting better results by setting a reasonable value for the fixed thickness. Relevant descriptions have been supplemented in Page 5, Paragraph 1, Line 3, and also in Supplementary Note 1 in SI.

4. *The authors defined a dimensionless IR camouflage evaluation factor RTS, which depends on the thermal equilibrium of the metamaterials and the ambient environment. Does it also work in an unknown or varying environment (e.g., the convective heat transfer coefficient, the ambient temperature)?*

Authors' Response:

Thanks for the question. In an unknown environment, the RTS could not work due to the lack of temperature information. However, the RTS can still work in a varying environment by applying the lumped parameter method.

The RTS can be calculated as follow:

$$RTS = \frac{\int_8^{13} \varepsilon(\lambda) E_{b\lambda}(T_1) d\lambda}{\int_8^{13} E_{b\lambda}(T_2) d\lambda}$$

Therefore, temperature information is needed to obtain the RTS. In a steady-state situation where the convective heat transfer coefficient and the ambient temperature are fixed, the camouflage metamaterial will achieve a fixed steady-state temperature eventually. The steady-state temperature can be calculated according to the thermal equilibrium equation as follow:

$$P_{inner} + P_{amb} = P_{conv} + P_{rad}$$

in which the inner heating power P_{inner} , the inward radiation power from the ambient P_{amb} , the outward heat convection power P_{conv} , and the outward radiation power P_{rad} are balanced.

In a varying environment, although the convective heat transfer coefficient or the ambient temperature does not remain constant, the transient lumped modeling of real-time temperature of the object can be considered as follow:

$$\rho c V \frac{\partial T}{\partial \tau} = P_{inner} + P_{amb} - P_{conv} - P_{rad}$$

where ρ , c , V are the density, specific heat, and volume of the object. The real-time temperature can be averaged and substituted into the first equation to obtain the RTS. Relevant discussion has been supplemented in Page 7, Line 8 from bottom and marked red.

5. Compared to the existing works (e.g., *ACS Appl. Nano Mater.* 2022, 5, 5119–5127; *ACS Appl. Mater. Interfaces* 2022, 14, 24690–24696) on visible-to-infrared camouflage, how much performance improvement can be achieved by the inverse design protocol than conventional empirical design strategies?

Authors' Response:

Thanks for the good question and the suggestion on the comparison with the two excellent work. The first research empirically designed a Ge/ZnS multilayer structure to fulfill a reflectance tuning in both visible and infrared. We reproduce the infrared reflectance of their proposed structure (shown in the Supporting Figure R3(a) below) and evaluate its RTS with the same environment settings. The accuracy of our simulated reflectance is verified by comparing with the counterpart in their text, which is shown in the Supporting Figure R3(b) below. The blue solid line which represents our reproduced IR reflectance matches well with the red dash line which represents the counterpart in their text, indicating the accuracy of our reproduced reflectance. The empirically designed structure exhibits an RTS of 0.5027, which means that it generates half as much thermal signal as the black body under the same conditions. Meanwhile, all of our optimized results exhibit RTS below 0.42, with the best one exhibit an RTS of 0.3471. Therefore, it is proved that our structure through optimized has better infrared camouflage than this structure with empirical design. In the second research, an oxalate-rich porous alumina film was used to achieve infrared camouflage, and the displayed color was modulated with different nanoparticle fillings. Since the nanoparticle materials and the composite porous alumina are not combined in a layered stacking way, their structural topology is different from the multi-layered topology studied in our research. Therefore, it is not suitable to be compared with our design.

In general, compared with the first research, our optimized structure can shield more than 30% of the thermal signal equivalent to the artificially designed structure, which is the performance improvement achieved by our inverse design protocol. Relevant discussion has been supplemented in Page 15, Paragraph 1, Line 11 from bottom and marked red. Details can be seen in Supplementary Note 5.

Supporting Figure R3 (also Supplementary Figure S5) (a) Schematic of the structure proposed in *ACS Appl. Nano Mater.* 2022, 5, 5119–5127. (b) Reflectance spectra. Solid line: from our reproduced reflectance data; Dash line: from original data.

6. *There are some details that should be considered in a revision.*

In Fig 1b and Fig 2b, it is better to change the x label to “Iteration steps” to make it more precise.

In Fig 5b, the spacing of each row should be adjusted; the font size of Fig 5c is relatively too small and should be adjusted too.

The English writing should be better polished and some typos should be corrected.

Authors' Response:

Thanks for the suggestions and corrections. The mentioned figures have been adjusted and the English writing has been polished.

Other comments on some minor issues:

7. *Recently there are some new developments on thermal metamaterials. The authors are suggested to cite more references. To list a few:*

Li, Y. et al. Transforming heat transfer with thermal metamaterials and devices. Nat. Rev. Mater. 6, 488-507 (2021).

Xu, G. et al. Diffusive topological transport in spatiotemporal thermal lattices. Nat. Phys. 18, 450-456 (2022).

Li, J. et al. Reciprocity of thermal diffusion in time-modulated systems. Nat. Commun. 13, 1-8 (2022).

Xu, L. et al. Thermal Willis Coupling in Spatiotemporal Diffusive Metamaterials. Phys. Rev. Lett. 129, 155901 (2022).

Authors' Response:

Thanks for the good suggestions. We have supplemented some relevant references, which have been marked as red.

Responses to the comments of the Second Reviewer:

Reviewer #2: In the manuscript entitled “Ultra-high-efficient material informatics inverse design of thermal metamaterials for visible-infrared-compatible camouflage” by Wang Xi et al, a Bayesian optimization algorithm coupled with transfer matrix method is employed to design multilayer germanium (Ge) and zinc sulfide (ZnS) metamaterials for VIS-IR camouflage. They experimentally fabricated the Ge/ZnS multilayer metamaterial atop quartz substrate and measured the visible reflectance and IR emissivity. The experimental results agree well with numerical calculations. This is a solid demonstration, but it lacks the novelty to be published in top journals like Nature Communications. In fact, Zhu et al have experimentally demonstrated multispectral camouflage for the visible, mid-infrared (MIR, 3–5 and 8–14 μm), lasers (1.55 and 10.6 μm) and microwave (8–12 GHz) bands with simultaneous efficient radiative cooling in the non-atmospheric window (5–8 μm) [Zhu, Huanzheng, et al. "Multispectral camouflage for infrared, visible, lasers and microwave with radiative cooling." *Nature communications* 12.1 (2021): 1-8]. Their device consists of a ZnS/Ge multilayer for wavelength selective emission and a Cu-ITO-Cu metasurface for microwave absorption. Their design is more sophisticated, more powerful compared to the design here in this manuscript where a similar ZnS/Ge multilayer was employed. At the same time, Bayesian optimization has also previously been employed for the design of metamaterials [see e.g, Sakurai, Atsushi, et al. "Ultrannarrow-band wavelength-selective thermal emission with aperiodic multilayered metamaterials designed by Bayesian optimization." *ACS central science* 5.2 (2019): 319-326]. As a result, I have to suggest the rejection of this manuscript.

Authors' Response:

Thanks for your valuable comments. Regarding to your main concern about the novelty compared to Zhu et al. and Sakurai et al, we insist the novelty as follows.

1. *Advance to [Zhu, Huanzheng, et al. "Multispectral camouflage for infrared, visible, lasers and microwave with radiative cooling." *Nature communications* 12.1 (2021): 1-8]*

Zhu et al. empirically designed a coupled structure (shown in the Supporting Figure R4(a) below), which consisted of a Ge/ZnS multilayer and a Cu-ITO-Cu metasurface, to demonstrate multispectral camouflage for the visible, mid-infrared (MIR, 3–5 and 8–14 μm), lasers (1.55 and 10.6 μm) and microwave (8–12 GHz) bands. They overcame the challenge of several spectral requirements with one structure and achieved a satisfying compatible camouflage effect, which is quite difficult and requires expert intuition.

In contrast, our research proposes a material-informatics-based inverse design framework to efficiently design metamaterials with desired camouflage target, which avoids the difficult and complicated empirical design process. It should be noted that although the target is set as VIS-IR compatible camouflage in our text as an example, the inverse design framework also applies to more complicated situation such as VIS-IR-laser-microwave-compatible camouflage after modifying the optimization target, structure pattern, alternative materials, simulation method, and so on.

Our second advantage over Zhu's work is that our results demonstrate more colors and equally good IR camouflage effect with simpler structure. For verification, the colors obtained in Zhu's

work has been marked as black square in the CIE 1931 color space (Supporting Figure R4(c)) below. It can be seen that these colors are included in the purple dashed line, which shows that the color range in our design is larger. The IR reflectance of part of their stacked structure (only the part which contributes to VIS-IR camouflage) has been also reproduced with its *RTS* evaluated in the same ambient condition. The accuracy of the result is verified by comparing with the IR reflectance in their paper, which is shown in the Supporting Figure R4(b) below. The *RTS* is calculated to be 0.3479, which is on par with our optimized structure (*RTS*=0.3471). Nevertheless, their structure contains eleven alternating Ge/ZnS layers whose thicknesses are 0.721/0.982/0.721/0.559/0.234/0.438/0.206/0.438/0.552/1.18/0.701 μm (from top to bottom), which is more complicated and harder to manufacture compared to our three-layer structure.

In conclusion, our widely applicable inverse design framework can free us from empirical design with thresholds to explore better performance and simpler structures, which may provide reference and spark new ideas for multi-objective optimization including multi-spectral camouflage. The comparison between Zhu's work and ours are listed below in Table T1.

Supporting Figure R4 (also Supplementary Figure S1) (a) Schematic of the structure proposed in *Nat. Commun.* 2021, 12, 1805. (b) Reflectance spectra. Solid line: from our reproduced reflectance data; Dash line: from original data. (c) CIE 1931 color space. Purple cross: structures designed by our inverse design algorithm; Purple dotted line: the color range of our structures; Black squares: the structures in *Nat. Commun.* 2021, 12, 1805.

Table T1 Comparison between Zhu's structure and ours

	Zhu's structure	Our structure
Design Method	Empirically Designed	Optimized Automatically
Number of Total Layers	11 layers	3 layers
Total thickness	6.732 μm	~2.340 μm

Color Range		
IR Camouflage Performance (A smaller RTS means a better performance)	RTS =0.3479	RTS =0.3471

2. Advance to [Sakurai, Atsushi, et al. "Ultraviolet-narrow-band wavelength-selective thermal emission with aperiodic multilayered metamaterials designed by Bayesian optimization." *ACS central science* 5.2 (2019): 319-326]

Thanks for suggesting Prof. Junichiro Shiomi's paper for improving our manuscript.

Actually, the last author (Prof. Run Hu) is a previous postdoc of Prof. Junichiro Shiomi at The University of Tokyo, and learned the Bayesian optimization algorithm there. We have co-authored many papers like PRX and Adv. Mater. We applied Bayesian optimization to accelerate the design of aperiodic GaAs/AlAs superlattice to minimize coherent phonon heat conduction with revealing the physics and experimental validation (*Phys. Rev. X* 10, 021050, 2020). After back to China, we applied such algorithm to design thermal emitter to maximize the thermophotovoltaic performance (*Nano Energy* 72, 104687, 2020) or infrared sensing (*Opt. Lett.* 46, 888, 2021).

After receiving your review comments, we also send it to discuss with Prof. Junichiro Shiomi. Prof. Shiomi encouraged us to continue polishing our manuscript to outline the novelty. At the same time, he also admits that his group is also applying Bayesian optimization or other machine-learning algorithms for solving different problems, such as phonon nanostructures (*Phys. Rev. X* 7, 021024, 2017; *Phys. Rev. X* 10, 021050, 2020), thermoelectric materials (*Sci. Adv.* 4, eaar4192, 2018; *Mater. Horizon.* 8, 2463, 2021; *ACS Comb. Sci.* 22, 782, 2020), metamaterials (*ACS Cent. Sci.* 5, 319, 2019; *Phys. Rev. Research* 2, 013319, 2020), thermophotovoltaic emitter (*Nano Lett.* 2023), thermally conductive polymers (*Npj Comput. Mater.* 5, 66, 2019; *Mater. Today Phys.* 28, 100850, 2022), radiative cooling (*JHMT* 195, 123193, 2022), high-k crystals (*Phys. Rev. Mater.* 5, 053801, 2021), tunnel magnetoresistance (*Phys. Rev. Research* 2, 023187, 2020), etc.

Prof. Shiomi also welcomes others scholars to apply this algorithm that is originated from his group to solve different problem. After discussing with Prof. Junichiro Shiomi, we also feel that although we apply Bayesian optimization algorithm, we are not just taking this algorithm as our novelty of this manuscript.

Our work advances in following two aspects.

First, we have considered the influence of fabrication error by introducing the *SET* (Sensitivity of Emissivity to Thickness) factor into our objective function. In their work, the thickness error of fabricated samples leads to differences between their emissivity spectrum and the simulation results. We have learned from this and taken the *SET* of structure into consideration. At a certain level of fabrication error, a smaller *SET* of the structure means a better match with the emissivity of fabricated sample and the simulation result. It is such configuration in optimization that helps all of our fabricated samples exhibit nearly the same emissivity spectra as the simulations.

Second, our work is a multi-objective optimization which have considered the color difference, the IR camouflage performance, and the fabrication error simultaneously, which is more comprehensive than theirs in the ACS paper. The comparison between Shiomi's work and ours are listed below in Table T2.

In short and in conclusion, our novelty includes: 1) solving the photonic structure design for colored infrared camouflage with quite good performance over the existing literatures in terms of large color space range and low *RTS* values; 2) demonstrating a general design roadmap for multi-objective optimization based on Bayesian algorithm and taking fabrication error into consideration for the first time, which can further be extended for other problem or more complicated structure design.

Table T2 Comparison between Shiomi's work and ours

	Prof. Junichiro Shiomi's work	Our work
Objective Function	$FOM = \frac{\%}{\varepsilon_{\lambda_i \pm \Delta\lambda}} - \frac{\%}{\varepsilon_{else}}$ which is single-objective optimization.	$FOM = RTS \times \Delta E_{CMC(l;c)} \times SET$ which is multi-objective optimization. RTS accounts for IR camouflage performance, $\Delta E_{CMC(l;c)}$ accounts for visible camouflage performance, TS accounts for machining error.
Is Fabrication Error Considered	No	Yes
Wavelength range	4-6 μm	0.38-17 μm

Comparison between simulation and experiment results

Responses to the comments of the Third Reviewer:

Reviewer #3: The paper titled "ultrahigh-efficient material informatics inverse design of thermal metamaterials for visible-infrared-compatible camouflage" uses multilayer Ge and ZnS metamaterials to achieve camouflage in visible and infrared bands at the same time. The authors harness Bayesian optimization to inverse design the distribution of Ge and ZnS in order to achieve targeted reflectance spectra for different colors. In my opinion, this work is interesting, complete, and potentially useful. However, I do have a few major comments from the material design and inverse design algorithm perspectives.

Authors' Response:

Thanks very much for your positive comments.

1. *In this paper, the design space is defined as follows: a topmost ZnS sublayer with a thickness from 20 to 600 nm, and a total number of 10 sublayers which can be either ZnS or Ge. To me, the selection of such design space is very random and seemingly requires a lot of expert intuition. Can authors explain more about why we should focus on this particular design space? Two questions need to be addressed here: a) Is the design space expressive enough, i.e., can the metasurface cover all/most of the colors and IR? b) Is the design space concise enough, i.e., how many designs will result in meaningful metasurfaces within this design space? In my opinion, the selection of the "design space" is the most crucial part of this paper. With a well-chosen design space, most algorithms can easily perform the optimization task. More justification and physical intuition needs to be added for this point.*

Authors' Response:

Thanks for your questions.

Our design space is actually a tradeoff between making the design space expressive enough and making the optimization feasible. A fully expressive design space can be achieved by designing an m layers structure where each layer has n choices. Suppose a five-layer structure ($m=5$), and each layer thickness ranges from 20 nm to 1 μm with a spacing of 10 nm with two kinds of material choices ($n=99\times 2$). Therefore, the design space is $(99\times 2)^5 = 3.04\times 10^{11}$, which requires 10^7 order of computing hour which is too large to solve. To make it solvable, we shrink the design space by three steps.

First, a single topmost ZnS layer with tunable thickness is reserved to achieve color. Several papers have proved that for layered structure, the whole can be divided into two parts which corresponds to the VIS and IR emissivity respectively. For example, Qi et al. empirically designed a Ge/ZnS layered structure to achieve IR camouflage with color in *Opt. Lett.* 43, 21 (2018): 5323-5326. The topmost ZnS corresponds to color tuning while the rest part corresponds to IR camouflage. Sheng et al. designed a top SiO₂/SiN emitter for IR emissivity tuning and a bottom MgF₂/SiC Tamm structure for VIS emissivity tuning in *ACS Photonics* 2019, 6, 2545–2552. Here, only one topmost layer other than more layers is chosen to achieve color for simplification. After preliminary study, the thickness range of the topmost ZnS sublayer is set as 20~600 nm with a spacing of 10 nm, causing 59 choices.

Second, the sublayer thickness of the m -layer structure for IR camouflage is fixed, which reduce

n to 2 (two alternative materials). Since each sublayer only has two choices, the binary digits of 0 and 1 can be used to represent the structure. Although the limited searching space could not guarantee the globally optimal result, we can increase the probability of finding it by setting a reasonable value to the fixed sublayer thickness. In our research, the thickness is set to be 200 nm, which ensures the opportunity to form Ge/ZnS pair, whose optical path is approximate to the quarter of target wavelength in the IR band to induce more constructive interferences.

Last, the number of sublayers is loosened to 10 ($m=10$), which can cause the design space in an order of 10^4 ($59 \times 2^{10} = 6 \times 10^4$) and makes the whole optimization time-friendly.

As for the first mentioned question, our proposed pattern is only expressive for the layered pattern. Structures in other patterns (such as grating pattern, nanorod pattern, etc.), which can possibly exhibit more colors or better infrared camouflage performance, are not included in our design roadmap. Nevertheless, the roadmap offers a universal design approach, which can work again after necessary modifications including structure pattern, design space and simulation method. For the second mentioned question, all structures in our design space are meaningful and manufacturable since there is no further constraint in our structure pattern. Even structures with poor performance will help the overall optimization evolves in the right direction under the Bayesian algorithm.

Relevant descriptions have been supplemented in Page 5, Paragraph 1, Line 3 and also in Supplementary Note 1 in SI.

2. *The authors mentioned in the introduction that "... these metamaterial designs are mainly dependent on conventional empirical design strategies or manual optimization, and thus their performances are far from that of the globally optimized one". However, there is no guarantee that the designs discovered by the authors are globally optimal. Such sub-optimal is two-fold: a) Since Bayesian optimization (BO) only evaluated 1% of the design space, it is unknown how good the design discovered by BO compared to the global optimal. b) the total 60416 designs only represent a small fraction of realizable designs. The optimal design among these 60416 designs is neither globally optimal. Therefore, I think the authors should be careful when using the word "optimal".*

Authors' Response:

Thanks for your suggestion. We apologize for the careless use of the word "globally optimal" and have modified our words to avoid misleading. Actually, the "optimal structure" in our text is referred to the optimal one among the 60416 cases in our design space. Supporting Figure R5(c) demonstrates that when calculating 1% of 60416 cases, we obtain the optimal structure; while continuing calculating 30% of total cases, the optimal structure is not changed. Therefore, it can be inferred that our proposed structure is the optimal one in our design space. Although it is not the globally optimal one, it still exhibits better infrared camouflage performance than those with empirical design. For example, Deng et al. empirically designed a Ge/ZnS multilayer structure (shown in the Supporting Figure R5(a) below) to fulfill a reflectance tuning in both visible and infrared [*ACS Appl. Nano Mater.* 2022, 5, 5119–5127]. We reproduce their infrared reflectance and evaluate its *RTS* in the same ambient condition. The accuracy of our simulated

infrared reflectance is verified by comparing with the counterpart in their text, which is shown in the Supporting Figure R5(b) below. The RTS is calculated to be 0.5027, which means that it generates half as much thermal signal as the black body under the same conditions. Meanwhile, all of our optimized results exhibit RTS below 0.42, with the best one exhibit an RTS of 0.3471. Our optimized structure can shield more than 20% of the thermal signal equivalent to the artificially designed structure. Relevant descriptions about the “globally optimized” in text have been modified and marked red.

Supporting Figure R5 (also Supplementary Figure S6) (a) Schematic of the structure proposed in *ACS Appl. Nano Mater.* 2022, 5, 5119–5127. (b) Reflectance spectra. Solid line: from our reproduced reflectance data; Dash line: from original data. (c) FOM evolution with iteration step. Inset: Color display and emissivity spectrum of four intermediate structures during the optimization.

3. The authors compared two FOMs, i.e., FOM_1 and FOM_2 , where $FOM_2 = RTS$. It seems that the VIS camouflage requirement does not affect IR camouflage performance much (only a few percent inferior). Is this true for all VIS camouflage colors? If this is true for all other colors, it may indicate that the camouflage requirements for VIS and IR might be decoupled in the current design space. From these observations, the authors might find some physical explanations for why such a design space works well.

Authors' Response:

Thanks for your suggestion. Although VIS camouflage requirement has little influence on IR camouflage performance, we still optimize in a coupled way considering the universality and

extensibility of our inverse design algorithm. The reason why VIS camouflage has little effect on infrared camouflage is that the energy proportion of thermal radiation in visible band is very small. According to our calculation, our designed structures with infrared camouflage effect maintain temperature of 320~420K, whose thermal radiation mainly concentrates within 5~13 μ m. However, the emissivity requirement of VIS camouflage is only in 0.38~0.78 μ m, which has little influence on the heat balance of object, therefore interferes little on infrared camouflage. Nevertheless, whether they can be optimized separately also depends on the specific structure. For layered structure, the whole can be divided into two parts, which respectively relating to VIS and IR camouflage. For example, Qi et al. empirically designed a Ge/ZnS layered structure to achieve IR camouflage with color [Dong, Qi et al. *Effective strategy for visible-infrared compatible camouflage: surface graphical one-dimensional photonic crystal*, *Optics Letter* 43, 21 (2018): 5323-5326]. The topmost ZnS corresponds to color tuning while the rest part corresponds to IR camouflage. Sheng et al. designed a top SiO₂/SiN emitter for IR emissivity tuning and a bottom MgF₂/SiC Tamm structure for VIS emissivity tuning [Chunxiang Sheng et al. *Colored Radiative Cooler under Optical Tamm Resonance*, *ACS Photonics* 2019, 6, 2545–2552]. However, for other structures (such as grating), it is not clear which part of the structure corresponds to VIS or IR emissivity. In addition, if the grating width mainly affects VIS and the grating period affects IR, then the structural constraints such as “grating width < period” should be considered when optimize decoupled. Therefore, coupling optimization not only does not need to clarify the correspondence relationship with structure and spectrum, but also saves the extra work of integrating the two parts of the results obtained by decoupled optimization. Therefore, considering the universality and extensibility of the design process, we still adopt the coupling optimization scheme in this layered structure situation. Relevant discussions have been supplemented in Page 11, Paragraph 1, Line 4 and marked red.

4. *I think the authors should also come up with some criteria characterizing the difference between targeted color and inverse-designed color. Such criteria should be used to measure both the simulation error and the experimental error. A photograph can be used to determine the color of an experimentally fabricated sample.*

Authors' Response:

Thanks for your suggestion. The photograph of the experimentally fabricated samples has been shown in Fig 3(c). The color difference $\Delta E_{CM(l;c)}$ has already been used to measure the simulation error, which is the difference between the target color and the inverse-designed color. Meanwhile, the experimental error, which is the difference between the target color and the fabricated color, can also be characterized by $\Delta E'_{CM(l;c)}$. We have supplemented the simulation error and experimental error by labelling them in Fig 3 (as shown below in Supplementary Figure R6). Although the experimental color differences are larger than the simulation counterpart, the reflectivity spectra and chroma and hue are similar. It should be noted that the simulation color difference represents how excellent the design is in color matching, while the experimental color difference superimposes not only an unpredictable fabrication error but also the lighting condition inconsistency on top of it. Relevant adjustments have also been modified in section ‘Visible Camouflage Performance’ and marked red.

Supporting Figure R6 (also Figure 3) Color performance of both designed structures and fabricated samples. (a) The color range and distribution of simulated and fabricated structures in CIE 1931 color space. (b) Simulated and measured reflectivity spectra of eight structures in visible waveband with their target color, designed color and their color difference. (c) Photos of fabricated structures with their target color, fabricated color and their color difference.

Modification List

In Main Text:

1. Page 1, Line 10 in section 'Abstract', Modification: 'The informatic encoding ... with ~1% of the total candidate calculation steps.'
2. Page 3, Paragraph 1, Line 5, Modification: 'More sophisticatedly, the emissivity spectrum ... should stay high to cool the target's temperature through radiative cooling.'
3. Page 3, Paragraph 1, Line 11, Modification: 'since regulating the reflected properties ... intervene emissive properties in the IR bands, and vice versa.'
4. Page 3, Paragraph 1, Line 6 from bottom, Delete the sentence 'and thus their performances are far from that of the globally optimized one' after 'Nevertheless, these metamaterial designs ... or manual optimization'.
5. Page 3, Paragraph 2, Line 1, Modification: 'In this paper, we present a general material-informatics-based²⁴⁻²⁵ (MI) framework ... sophisticated reflectivity/emissivity spectra for VIS-IR camouflage.'
6. Page 4, Line 3 in section 'Roadmap of Metamaterial Design', Supplementary: 'both visible and IR camouflage evaluations as well as sensitivity of emissivity on thickness evaluation'
7. Page 5, Paragraph 1, Line 3, Supplementary: 'More details of how such structure configuration is set up can be seen in Supplementary Note 1'.
8. Page 5, Paragraph 1, Line 6, Modification: 'With such material and structure configuration, the MI-based inverse design process is implemented as follows'.
9. Page 5, Paragraph 2, Line 4 from bottom, Modification: 'The comparison of the two methods is provided ... to obtain the reflectance and the emittance spectra.'
10. Page 5, Paragraph 3, Line 1, Modification: 'Second, evaluation. We evaluate ... thickness sensitivity performance.'
11. Page 6, Paragraph 1, Line 3 from bottom, Modification: 'More details of the color characterization ... are presented in Supplementary Note 3'.
12. Page 7, Paragraph 1, Line 4, Modification: 'and specifically'.
13. Page 7, Paragraph 1, Line 13, Supplementary: 'and then evaluate the corresponding IR camouflage performance RTS according to Eq. (2)'.
14. Page 7, Line 9 from bottom, till Page 8, Line 7 from bottom. A Long Supplementary: 'It should be noted that in a varying environment ... which is preferred and ready for optimization.'
15. Page 9, Line 9 from bottom, Supplementary: 'No better result other than this optimal result in our design space which consists of 60416 designs is found till 30% of total candidates have been simulated'.
16. Page 10, Figure 1, Modification.
17. Page 10, Line 3 from bottom, Modification: 'we erect another round ... as shown in Figure 2a'.
18. Page 11, Paragraph 1, Line 4, Modification: 'Taking $RTS \times SET$ as the FOM_2 ... and IR camouflage performance RTS '.
19. Page 11, Paragraph 1, Line 6 from bottom, Modification: 'It is seen that taking $RTS \times SET$ as the FOM_2 ... the metamaterial is in green color which can be applied for VIS-IR camouflage.'

20. Page 11, Paragraph 2, Line 1, Supplementary: ‘It should be noted that if more topmost layers are introduced ... with grating, rods, square pattern, etc’.
21. Page 12, Figure 2, Modification. The description of figure has also been modified.
22. Page 13, Paragraph 1, Line 3, Modification: ‘The purple dashed line is the envelope curve encircling all the simulated structures’
23. Page 13, Paragraph 1, Line 8, Modification: ‘the present optimization algorithm can also be implemented, though outputting a similar color or a weakened IR camouflage performance’
24. Page 13, Line 13, till Page 14, Line 5, A Long Supplementary & Modification: ‘It should be noted that the color range can be expanded ... Compared to the simulation error, the experimental error originates from both the unpredictable fabrication error and the error caused by different lighting condition.’.
25. Page 14, Figure 3, Modification. The description of figure has also been modified.
26. Page 15, Line 12 in section ‘Infrared Camouflage Performance’, Supplementary & Modification: ‘Comparison of the infrared camouflage ...For further verification ... No. 8 ... are plotted in Figures 4i-l’.
27. Page 15, Paragraph 2, Line 2 from bottom, Modification: ‘The eight fabricated samples with different colors are settled ... which can display normal thermal signal under the same temperature’.
28. Page 16, Paragraph 1, Line 6, Modification & Deletion: ‘the fabricated samples ought to display lower temperature due to their IR camouflage function. The angle insensitive property can be verified by taking IR images in different observing angle.’
29. Page 16, Paragraph 1, Line 6 from bottom, Modification: ‘The angle-independent property can be further demonstrated ... The measured infrared emissivity spectrum of the No. 8 sample under different incident angles are plotted ... implying a rather good angle-independent IR camouflage performance’.
30. Page 16, Paragraph 2, Line 3, Supplementary: ‘Our work develops an ultrahigh-efficient machine-learning-kernel inverse design framework ... More detailed discussions can be seen in Supplementary Note 7.’
31. Page 17, Figure 4, Modification. The description of figure has also been modified.
32. Page 17, Figure 5, Modification. The description of figure has also been modified.
33. Page 18, Line 4 in section ‘Conclusion’, Modification: ‘but also obtaining broad color and IR signals with background-matching characteristics.’.
34. Page 18, Line 7 in section ‘Conclusion’, Supplementary: ‘evaluate the VIS-IR camouflage and the sensitivity of emissivity on thickness.’.
35. Page 18, Line 9 in section ‘Conclusion’, Modification: ‘With iteration steps less than 1% ... The present VIS-IR metamaterials exhibit promising potential in multispectral camouflage applications.’.
36. Seven references have been supplemented and the reference sequences have been updated.

In Supplementary Information

1. Section Supplementary Note 1, 4, 5, 7 have been added. The sequence of supplementary notes has been updated.

2. Supplementary Figure S3, S4, S6 have been added. The sequence of supplementary figures has been updated.

Reviewer #1 (Remarks to the Author):

i have gone through the response and revised version. most of my previously raised concerns have been well addressed. I thereby recommend the acceptance.

Reviewer #2 (Remarks to the Author):

I have read the author rebuttal letter and I still believe that the results of this paper are incremental.

(1) Even though inverse design was employed, the optimization is local and the design still relies on expert intuition. In fact, this paper uses a typical Ge/ZnS multilayer as in Zhu's paper[Zhu, Huanzheng, et al. "Multispectral camouflage for infrared, visible, lasers and microwave with radiative cooling." Nature communications 12.1 (2021): 1-8].

(2) As I mentioned, the results in this paper don't show much improvement compared to Zhu's work. Even though this paper demonstrates more colors, the color range in this paper isn't much wider than Zhu's work. In fact, I believe similar colors can also be realized with Zhu's design by varying the geometric parameters.

I think the results are publishable but cannot meet the novelty of Nature Communications.

Reviewer #3 (Remarks to the Author):

The authors have addressed all my previous comments. I appreciate the time and effort the authors put in the revision process. I do not have further comments and recommend the publication of this work.

Response to Reviewers' comments

Responses to the comments of the First Reviewer:

Reviewer #1: I have gone through the response and revised version. Most of my previously raised concerns have been well addressed. I thereby recommend the acceptance.

Authors' Response:

Thanks very much for your recommendation.

Responses to the comments of the Second Reviewer:

Reviewer #2: I have read the author rebuttal letter and I still believe that the results of this paper are incremental. I think the results are publishable but cannot meet the novelty of Nature Communications.

1. *Even though inverse design was employed, the optimization is local and the design still relies on expert intuition. In fact, this paper uses a typical Ge/ZnS multilayer as in Zhu's paper[Zhu, Huanzheng, et al. "Multispectral camouflage for infrared, visible, lasers and microwave with radiative cooling." Nature communications 12.1 (2021): 1-8].*

Authors' Response:

Thanks for your valuable comments. Your concerns about the novelty of our design process compared to Zhu's work are expressed in mainly two aspects, which is using the same material and still relying on expert intuition.

Firstly, using the same materials does not diminish the novelty of our work. Ge and ZnS is chosen due to their relatively large difference in the IR refractive index, which can be used to effectively tune the emissivity spectra in the target IR waveband. Actually, the two materials are commonly used for emissivity tuning in visible and infrared waveband and have shown up in many other studies, such as *Infrared Phys. Techn.* 2016, 79, 144-150; *Opt. Lett.* 2018, 43(21), 5323-5326; *Materials.* 2018, 11(9), 1594; *Light Sci. Appl.* 2020, 9, 60, et al.

Secondly, as described in our structure configuration details, only a basic knowledge of optics is required in setting the thickness of the sublayers, that is, setting the thickness value so that optical path equals to one quarter of the target wavelength, thus increasing the reflectivity and reducing the emissivity. However, the structure proposed in Zhu's work has eleven layers, whose thicknesses are 0.721/0.982/0.721/0.559/0.234/0.438/0.206/0.438/0.552/1.18/0.701 μm (from top to bottom), which relies on expert intuition and rich design experience. Compared to Zhu's work, our design framework requires less manual design with simpler structures and better performance, hence facilitating the further exploration, extension, and application of the present design framework and algorithm.

2. *As I mentioned, the results in this paper don't show much improvement compared to Zhu's work. Even though this paper demonstrates more colors, the color range in this paper isn't much wider than Zhu's work. In fact, I believe similar colors can also be realized with Zhu's*

design by varying the geometric parameters.

Authors' Response:

Thanks for your valuable comments, which express your concern on the color range. Different from artificial design, our work adopts inverse design framework to automatically screen the feasible colors by filtering the massive structures in searching space without human supervision. With the help of inverse design framework, the feasible color range is expanded, which include the colors obtained in Zhu's work.

Furthermore, the color range shown in our text is just a demo since only one single topmost layer is employed to demonstrate color in our work. Our inverse design framework is extensible and enables an expansion of the feasible color range by applying other structure patterns to demonstrate color. For instance, we design three structures (α , β , γ) from the new proposed metasurface pattern and simulate their visible reflectance with RCWA (Rigorous Coupled Wave Analysis) to obtain their colors. The visible reflectance and colors are plotted in Supporting Figure R1(c) with the structure parameters labeled. The colors of three structures (α , β , γ) are marked as diamonds in CIE 1931 color space compared with the color range of the layered structures, which is lined out by the purple dotted line, as shown in Supporting Figure R1(b). It can be seen that the colors of our new proposed structures lie beyond the purple dotted line, which indicates an expansion of color range. It should be noted that the proposed structure here is just a demo. In fact, the 35-nm-thick Ge layer and the 100-nm-thick Ag layer which is fixed here for simplification can also be tunable. Moreover, grating, rods, rectangles, etc. can be designed as new structure patterns with more tunable parameters for more colors. It can be inferred that there are great chance to expand the color range after integrating such more complicated structure pattern in our inverse design framework, which may be explored in our near future work.

Supporting Figure R1 (also Supplementary Figure S2) (a) Schematic of the new proposed structure pattern. (b) CIE 1931 color space. Purple crosses: layered structures designed by our inverse design algorithm; Purple dotted line: the color range of the layered structures; Black

diamonds: the structures with new proposed pattern. (c) Visible reflectance of the three new proposed structures. Corresponding structure parameters and the colors are labelled.

Responses to the comments of the Third Reviewer:

Reviewer #3: The authors have addressed all my previous comments. I appreciate the time and effort the authors put in the revision process. I do not have further comments and recommend the publication of this work.

Authors' Response:

Thanks very much for your recommendation of our work.